# Enhanced Nanogel Formulation Combining the Natural Photosensitizer Curcumin and *Pectis brevipedunculata* (Asteraceae) Essential Oil for Synergistic Daylight Photodynamic Therapy in Leishmaniasis Treatment

**DOI:** 10.3390/pharmaceutics17030286

**Published:** 2025-02-21

**Authors:** Lara Maria Oliveira Campos, Estela Mesquita Marques, Daniele Stéfanie Sara Lopes Lera-Nonose, Maria Julia Schiavon Gonçalves, Maria Valdrinez Campana Lonardoni, Glécilla Colombelli de Souza Nunes, Gustavo Braga, Renato Sonchini Gonçalves

**Affiliations:** 1Laboratory of Chemistry of Natural Products, Department of Chemistry, Federal University of Maranhão (UFMA), São Luís 65080-805, Brazil; lara.moc@discente.ufma.br (L.M.O.C.); estela.marques@discente.ufma.br (E.M.M.); 2Department of Clinical Analysis and Biomedicine, State University of Maringá (UEM), Maringá 87020-900, Brazil; dssllnonose2@uem.br (D.S.S.L.L.-N.); pg405366@uem.br (M.J.S.G.); mvclonrdoni@uem.br (M.V.C.L.); 3Research Nucleus in Pharmaceutical Sciences Program, State University of Maringá (UEM), Maringá 87020-900, Brazil; gcsnunes2@uem.br; 4University College (COLUN), Federal University of Maranhão (UFMA), São Luís 65080-805, Brazil; gustavo.braga@ufma.br

**Keywords:** leishmaniasis, *Pectis brevipedunculata* essential oil, curcumin

## Abstract

**Background/Objectives**: Neglected tropical diseases (NTDs), such as leishmaniasis, remain a global health challenge due to limited therapeutic options and rising drug resistance. In this study, we developed an advanced nanogel formulation incorporating curcumin (CUR) and *Pectis brevipedunculata* essential oil (EO*Pb*) within an F127/Carbopol 974P matrix to enhance bioavailability and therapeutic efficacy against *Leishmania (Leishmania) amazonensis* (*LLa*) promastigotes. **Methods**: The chemical profile of EO*Pb* was determined through GC-MS and NMR analyses, confirming the presence of key bioactive monoterpenes such as neral, geranial, α-pinene, and limonene. The nanogel formulation (nG*P*C) was optimized to ensure thermosensitivity, and stability, exhibiting a sol–gel transition at physiological temperatures. Rheological analysis revealed that nG*P*C exhibited Newtonian behavior at 5 °C, transitioning to shear-thinning and thixotropic characteristics at 25 and 32 °C, respectively. This behavior facilitates its application and controlled drug release, making it ideal for topical formulations. Dynamic light scattering (DLS) analysis demonstrated that nG*P*C maintained a stable nanoscale structure with hydrodynamic radius below 300 nm, while Fourier-transform infrared spectroscopy (FTIR) confirmed strong molecular interactions between EO*Pb*, CUR, and the polymer matrix. Biological assays demonstrated that nG*P*C significantly enhanced anti-promastigote activity compared to free CUR and OE*P*b. **Results**: At the highest tested concentration (50 μg/mL EO*Pb* and 17.5 μg/mL CUR) nG*P*C induced over 88% mortality in *LLa* promastigotes across 24, 48, and 72 h, indicating sustained efficacy. Even at lower concentrations, nG*P*C retained dose-dependent activity, suggesting a synergistic effect between CUR and EO*Pb*. These findings highlight the potential of nG*P*C as an innovative nanocarrier for daylight photodynamic therapy (dPDT) in the treatment of leishmaniasis. Future studies will investigate the underlying mechanisms of this synergism and explore the potential application of photodynamic therapy (PDT) to further enhance therapeutic outcomes.

## 1. Introduction

Neglected tropical diseases (NTDs) represent a critical global health challenge, disproportionately affecting populations in tropical and subtropical regions, often living under conditions of extreme poverty [1]. Over one billion individuals are estimated to be impacted by NTDs, including conditions such as leishmaniasis, Chagas disease, and lymphatic filariasis [2]. These diseases are associated with significant morbidity, prolonged disability, and substantial economic costs. Despite their widespread prevalence and the immense burden they impose, NTDs remain largely neglected in the global health agenda, facing barriers such as limited funding, insufficient drug development, the rise in drug resistance, and challenges in vector control [3,4,5]. These complexities underscore the urgent need for novel therapeutic strategies to address conventional treatments’ limitations, particularly through innovative delivery systems that provide targeted, accessible, and effective solutions [6,7].

*Leishmania (Leishmania) amazonensis* (*LLa*) is one of the most prevalent protozoan parasites responsible for cutaneous leishmaniasis in the Americas, including Brazil, Peru, and other Latin American countries [8,9]. Transmitted by the bite of infected Lutzomyia, *LLa* primarily affects the skin, causing the formation of painful lesions or ulcers. If left untreated, these lesions can progress into chronic wounds, leading to irreversible scarring and, in some cases, mucocutaneous leishmaniasis [10]. The parasite’s ability to invade and replicate within macrophages, which are essential for immune defense, complicates the host’s immune response, allowing the infection to persist. While conventional therapies like pentavalent antimonials and miltefosine are available, their effectiveness is often limited by increasing drug resistance, side effects, and the need for prolonged treatment [10]. As a result, there is a pressing demand for alternative therapeutic strategies, particularly those leveraging advanced drug delivery systems such as nanogels, which can enhance drug efficacy and target the parasite more specifically [11,12,13].

Natural products, particularly essential oils (EOs) from plants native to the Amazon rainforest, have gained significant attention as promising therapeutic agents for NTDs. These oils, rich in bioactive molecules such as terpenoids and phenolic compounds, exhibit potent antimicrobial, antiparasitic, and anti-inflammatory properties, making them ideal candidates for addressing diseases like leishmaniasis [14,15]. The combination of EOs from different plant species or the incorporation of plant extracts into EO formulations has been shown to create synergistic or complementary effects, amplifying their therapeutic potential while minimizing the adverse side effects associated with conventional drugs. By utilizing the complex array of bioactive compounds in these natural products, researchers can develop more robust, effective, and targeted treatments for NTDs [16,17,18,19,20,21].

Accordingly, we hypothesize that a nanogel formulation based on the F127 copolymer and carbopol 974P, loaded with low concentrations of EO of *Pectis brevipedunculata* (EO*Pb*) and curcumin (CUR), will exhibit synergistic effects in vitro against *LLa* promastigote cells. *Pb* (Asteraceae), a plant native to the tropical and subtropical regions of the Americas, is known for its anti-inflammatory, antimicrobial, and antiparasitic properties [13,22]. The EO, characterized by a high content of monoterpenes like neral, geranial, α-pinene, and limonene, demonstrates significant antimicrobial properties, underscoring its potential as a natural resource for developing treatments against parasitic infections [23,24]. Previous studies from our group have demonstrated that EO*Pb*-loaded nanogels possess effective larvicidal, leishmanicidal, and anti-inflammatory activities, without cytotoxic effects [12,13]. In parallel, CUR, derived from *Curcuma longa*, is a well-established compound with proven leishmanicidal effects and has been extensively studied for its ability to inhibit parasite growth, modulate immune responses, and reduce inflammation. CUR’s low toxicity and antioxidant properties further enhance its potential as a therapeutic agent [25,26,27].

In this study, we designed a semi-solid nanoplatform following Green Chemistry (GC) principles, employing FDA-approved excipients while minimizing energy consumption [12,13]. A thermosensitive nanogel based on F127/974P was selected for EO*Pb*/CUR delivery due to its rapid sol–gel transition, stability, and suitability for topical applications. Unlike liposomes, micelles, or nanoemulsions, this system was developed through a low-energy, eco-friendly approach without the use of organic solvents. Its semi-solid nature enhances spreadability, bioadhesion, and skin permeation, making it an effective platform for cutaneous disease treatment [28]. To investigate its therapeutic potential, EO*Pb*/CUR was incorporated into the nanogel matrix at low concentrations and assessed for its efficacy against *LLa* promastigotes. The cytotoxicity of these compounds was specifically assessed at concentrations below 50 μg/mL of EO*Pb* and 20 μg/mL of CUR, both as isolated bioactive compounds and in combination within the nanogel. These concentrations were selected based on their biological activity against *LLa*, as reported in the literature on EO*Pb* and CUR [26,27,29]. Notably, while nanogels containing individual compounds (EO*Pb*- and CUR-loaded nanogels) exhibited limited activity, the combined formulation (EO*Pb*/CUR-loaded nanogel) demonstrated significant anti-promastigote effects, inducing dose-dependent mortality in *LLa* promastigotes. This study not only introduces a sustainable alternative for leishmanicidal therapies but also establishes nanogels as efficient carriers for dPDT without the need for high-energy light sources. The dPDT approach thus emerges as an environmentally friendly, cost-effective, and operationally viable strategy, paving the way for more efficient and sustainable therapeutic systems.

## 2. Materials and Methods

### 2.1. Materials

Carbopol 974P NF polymer was supplied by IMCD114 Brasil (São Paulo, SP, Brazil). Ultrapure water, sodium chloride, anhydrous sodium sulfate (≥99%), curcumin (≥99%), XTT (2,3-bis(2-methoxy-4-nitro-5-sulfophenyl)-5-[(phenylamino)carbonyl]-2H-tetrazolium hydroxide), PMS (N-methyl dibenzopyrazine methyl sulfate), penicillin, streptomycin, fetal bovine serum, and amphotericin B (AmB) were procured from Merck (Rahway, NJ, USA). Pluronic F127, a triblock copolymer of poly(ethylene oxide)-poly(propylene oxide)-poly(ethylene oxide) (MW = 12,600 g/mol; EO_99_PO_67_EO_99_), was also acquired from Merck.

### 2.2. Plant Material

The herbaceous species *Pb* was obtained from the Universidade Federal do Maranhão (UFMA) campus in São Luís, Maranhão, Brazil, at coordinates 2°33′20.5″ S and 44°18′32.7″ W. Following collection, a voucher specimen (No. 5287) was cataloged and preserved in the Rosa Mochel Herbarium (SLUI) at the Universidade Estadual do Maranhão (UEMA) in São Luís, MA, Brazil. The sampling adhered to Brazilian regulations for biodiversity protection and was officially recorded under SisGen code AAFB38B.

### 2.3. Extraction Procedure

The (EO*Pb*) was extracted through hydrodistillation using a Clevenger-type apparatus [12]. For this process, 300 g of air-dried plant material was finely cut with pruning shears to optimize extraction. The prepared material was then immersed in 500 mL of distilled water within a flask, and hydrodistillation was carried out for 2.5 h after reflux initiation. Following extraction, the obtained oil/water (O/W) mixture was centrifuged at 3500 rpm for 10 min at 25 °C. To remove any remaining moisture, the oil phase was dried over anhydrous sodium sulfate. The extraction yield of EO*Pb*, calculated relative to the dry weight of the plant material, was determined to be 0.81%.

### 2.4. CG-MS Analyses of EOPb

The chemical characterization of EO*Pb* was conducted using GC-MS and NMR techniques, following standardized methodologies [12]. For gas chromatography and mass spectrometry analyses, a Shimadzu system equipped with an RXi-1MS fused capillary column was employed, using helium as the carrier gas. The temperature gradient was carefully adjusted, and sample solutions (10 mg/mL in CH_2_Cl_2_) were injected with a 1:50 split ratio. Retention indices were established through a homologous series of n-alkanes, while peak areas and retention times were recorded to determine the relative composition of the constituents. GC-MS analyses were performed using a Shimadzu QP2010 SE system with an AOC-20i auto-injector under the same conditions applied in GC. Component identification was based on retention times, retention indices, and mass spectral comparison against reference libraries (ADAMS and FFNSC) and literature data.

For NMR spectroscopy, ^1^H, ^13^C, and DEPT-^13^C spectra were recorded using a BRUKER Avance III HD spectrometer (11.75 Tesla), operating at 500.13 MHz for ^1^H and 125.76 MHz for ^13^C. The samples were dissolved in deuterated chloroform (CDCl_3_), and chemical shifts were reported in ppm, with tetramethylsilane (TMS) as the internal reference.

### 2.5. Preparation of Nanogels

The nanostructured formulations were prepared according to the methodology described by Schmolka, using a cold procedure [12,13]. A portion of the F127 copolymer was slowly incorporated into distilled water and kept in an ice bath at 5–10 ºC. The solution was maintained under slow stirring to ensure the hydration of the F127 polymer chains. Subsequently, the 974P polymer was added in small portions until complete solubility was achieved, leading to the formation of the nanogel. Following this, the EO*Pb* was added, and the system was stirred for 30 min. To finalize the organization of the polymer chains, the final solution was refrigerated (5 ºC) overnight, resulting in the nanogel containing EO*Pb*. The preparation of the nanogel containing CUR followed the same methodology, except for the process of incorporating CUR molecules into the F127 copolymeric micelles, where a direct addition process was employed. In this case, a small amount of F127 was placed in a round-bottom flask containing distilled water at a temperature of 40 ºC with constant stirring. After solubilizing the F127, a quantity of CUR was added to the system and stirred at 40 ºC for 30 min. After cooling the system, the preparation process described for the nanogel containing EO*Pb* was followed to obtain the final material containing both EO*Pb* and CUR (nG*P*C) as shown in Figure 1A. The empty nanogel (nG) was prepared using the same steps, except the addition of EO*Pb* and/or CUR.

### 2.6. Stability Assay of Nanogels

To evaluate the influence of temperature on the physical and chemical stability of the nanogel formulations, accelerated stability studies were performed following ANVISA guidelines for cosmetics and the US Pharmacopeia [30]. Each formulation (1 mL) was subjected to centrifugation at 3000 rpm for 30 min at 25 ± 1 °C. The samples were then stored under two distinct conditions—ambient temperature (25 ± 3 °C) and refrigerated storage (5 ± 3 °C)—with continuous temperature monitoring throughout the experiment. Additionally, the formulations underwent a thermal stress test involving seven alternating cycles of 24 h at 5 °C followed by 24 h at 25 °C. Physical stability was assessed based on homogeneity, phase separation, and organoleptic properties, including visual appearance, color, and odor.

### 2.7. FTIR Analysis

FTIR analysis was carried out in reflectance mode using a Shimadzu FTIR Tracer-100 spectrophotometer (Kyoto, Japan). Lyophilized samples of nG, nG*P*, and nG*P*C were prepared as KBr pellets, whereas pure EO*Pb* was examined in Attenuated Total Reflectance (ATR) mode. ATR-FTIR measurements utilized a ZnSe crystal window (PIKE Technologies, Madison, WI, USA). Spectral data were recorded within the 400–4000 cm^−1^ range, maintaining a resolution of 8 cm^−1^ and averaging 50 scans per sample. To optimize spectral accuracy, samples were evenly spread on the ATR crystal surface, and the crystal window was meticulously cleaned with hexane and acetone before each measurement.

### 2.8. Scanning Electron Microscope (SEM)

The morphological characterization of nanogels nG and nG*P*C was conducted via SEM. The samples were initially flash-frozen in liquid nitrogen at −196 °C and subsequently lyophilized for 24 h using a Thermo Micro Modulyo freeze dryer (Thermo Electron Corporation, Pittsburgh, PA, USA). To improve imaging contrast, a fine metallic coating was deposited using a BAL-TEC SCD 050 Sputter Coater (Balzers, Liechtenstein). The morphology of the lyophilized samples was then examined at 100× and 50× magnifications using a FEI Quanta 250 microscope (Thermo Fisher Scientific, Karlsruhe, Germany).

### 2.9. DLS Analysis

The assessment of particle size, polydispersity index (PDI), and diffusional thermodynamic activation parameters was carried out through DLS using a Litesizer™ 500 analyzer (Anton Paar GmbH, Graz, Austria) equipped with a BM 10 module. The mean hydrodynamic radius ($R_{h}$) of nanogels nG and nG*P*C were measured in ultrapure water at four different temperatures: 25, 32, 37, and 45 °C. A 40 mW semiconductor laser operating at a 658 nm wavelength was employed for analysis. $R_{h}$ values were recorded using a 3.0 mL quartz cuvette. Each measurement was performed in triplicate, and results were expressed as mean ± standard deviation (SD). The diffusion coefficients ($D_{if}$) were calculated from the $R_{h}$ values using the Stokes–Einstein equation, assuming the formation of non-interacting spherical particles (Equation (Equation 1)) [31].(1)$$D_{i_{f}}=\frac{k_{B}T}{6\pi \eta R_{h}}$$
where $k_{B}$ is the Boltzmann constant ($1.3806503\times 10^{-23}$ J K^−1^), *T* is the absolute temperature (K), and η is the viscosity of the medium (Pa s^−1^). To obtain the values of the diffusional activation energy ($E_{\mathrm{ad}}$), as well as the diffusional entropy ($\Delta S_{d}^{‡}$) and enthalpy ($\Delta H_{d}^{‡}$), the Arrhenius and Eyring models, adapted for the diffusion process, were employed. The $E_{\mathrm{ad}}$ was determined using the following Arrhenius equation (Equation (Equation 2)) [32,33]:(2)$$D_{i_{f}}=Ae^{-\frac{E_{\mathrm{ad}}}{RT}}$$
where $D_{if}$ is the diffusion coefficient, *A* is the pre-exponential Arrhenius factor, *R* is the gas constant (8.314 J K^−1^ mol^−1^), and *T* is the absolute temperature in K. The plot of $\ln D_{i_{f}}=\ln A-\frac{E_{\mathrm{ad}}}{RT}$ gives $E_{\mathrm{ad}}$ by means of the slope. The calculations of $\Delta S_{d}^{‡}$ and $\Delta H_{d}^{‡}$ are obtained by the adapted Eyring model for diffusion (Equation (Equation 3)):(3)$$D_{i_{f}}=\frac{RT}{Nh}e^{\frac{\Delta S_{d}^{‡}}{R}}e^{-\frac{\Delta H_{d}^{‡}}{RT}}$$
where $N=6.02\times 10^{23}$ mol^−1^, and $h=6.626\times 10^{-34}$ J s. The plot of $\ln D_{i_{f}}=\ln\left(\frac{RT}{Nh}e^{\frac{\Delta S_{d}^{‡}}{R}}-\frac{\Delta H_{d}^{‡}}{RT}\right)$ gives the $\Delta H_{d}^{‡}$ using the angular coefficient, and the linear coefficient gives the $\Delta S_{d}^{‡}$. The enthalpy of activation ($\Delta H_{d}^{‡}$) is obtained from the slope of the plot of $\ln D_{i_{f}}T$, while the entropy of activation ($\Delta S_{d}^{‡}$) is obtained from the linear coefficient. Finally, the variation in the diffusive Gibbs free energy ($\Delta G_{d}^{‡}$) is calculated using Equation (Equation 4) [34,35]:(4)$$\Delta G_{d}^{‡}=\Delta H_{d}^{‡}-T\Delta S_{d}^{‡}$$

### 2.10. Rheological Analysis

The rheological behavior of nG*P*C was analyzed using a Thermo Scientific HAAKE MARS II controlled-stress rheometer (Waltham, MA, EUA). The instrument was configured with a parallel plate system, incorporating a titanium-coated steel cone (C35/2° Ti) with a diameter of 35 mm and a fixed gap of 0.105 mm to ensure precise measurements. All measurements were conducted at a controlled temperature of 5, 25, and 32 °C. Before analysis, the samples were equilibrated for 1 min to ensure thermal and structural stabilization. Flow curves were obtained by applying shear rates in two phases: an upward ramp from 0 to 500 s^−1^ and a downward ramp from 500 to 0 s^−1^. Each phase lasted 150 s, ensuring steady-state conditions were reached before transitioning between ramps. This protocol allowed the assessment of the shear-thinning behavior and any potential hysteresis effects in the nG*P*C.

### 2.11. In Vitro Assay Against LLa Promastigote Cells

*LLa* strain PH8 promastigotes were maintained in 199 culture medium supplemented with 10% heat-inactivated fetal bovine serum and antibiotics (100 UI/mL penicillin and 0.1 mg/mL streptomycin). Cultures were incubated at 27 °C and routinely subcultured to ensure parasite viability. For experimental assays, *LLa* promastigotes were seeded in 96-well plates containing RPMI 1640 medium, reaching a final concentration of 2 × 10^7^ parasites/mL after compound addition. The nG*P*C nanogel was prepared by incorporating 2 mg of EO*Pb* and 0.7 mg of CUR, subsequently diluted in RPMI 1640 medium. The final tested concentrations ranged from 50 to 6.25 μg/mL for EO*Pb* and from 17.5 to 2.19 μg/mL for CUR. For evaluating the nanogel containing the EO*Pb*/CUR combination, the same concentration range and dilution scheme were applied. The nG, nG*P*, and nGC were evaluated to assess the toxicity of the bioactive compounds in their non-combined forms, using the same concentration ranges.

An untreated *LLa* control was included to assess parasite viability, while AmB at 1.95 μg/mL served as a positive control. The plates were incubated at 27 °C for 24, 48, and 72 h to evaluate *LLa* promastigote viability. The colorimetric XTT assay (2,3-bis(2-methoxy-4-nitro-5-sulfophenyl)-5-[(phenylamino)carbonyl]-2H-tetrazolium hydroxide) was employed for this purpose. A reaction mixture containing 20% XTT, PMS (N-methyl dibenzopyrazine methyl sulfate), and 60% saline solution (0.9%) was added to each well. Following a 4 h incubation at 37 °C with 5% CO_2_, absorbance was measured at 450/620 nm using a spectrophotometer. The percentage of inhibition was calculated by comparing treated samples to untreated controls, with AmB as the reference drug.

All experiments were conducted in triplicate under conditions appropriate for dPDT, ensuring controlled light exposure parameters during the assay. Mortality rates were determined using logarithmic regression, based on a control curve generated from serial dilutions of Leishmania in culture medium, starting at 2 × 10^7^ parasites/mL and being progressively diluted by a factor of two until reaching 6.25 × 10^5^ parasites/mL.

### 2.12. Statistical Analysis

For the in vitro experiments, the half-maximal inhibitory concentration (IC_50_) of the tested compounds was calculated using logistic regression analysis based on the mean mortality at each tested concentration, employing Microsoft Excel (Microsoft Corporation, Redmond, WA, USA). All graphical representations were performed using Prism 9 (GraphPad, San Diego, CA, USA) and Origin Pro (version 8.5).

## 3. Results and Discussion

### 3.1. Development of Nanogels

Prior to the nanogel formulation, the chemical profile of EO*Pb* was characterized through GC-MS analysis, revealing a total of nineteen components, which account for 92.66% of the oil’s composition. Of these, 64.58% are oxygenated monoterpenes, represented by the isomers geranial (36.06%) and neral (28.52%), while the remaining compounds include α-pinene (15.72%) and limonene (8.28%). The chemical structures of the major compounds were confirmed through ^1^H and ^13^C NMR [13] and are represented in Figure 1B. According to previous studies, various combinations of F127 and 974P percentages were tested to find the optimal formulation that would allow the thermoresponsive behavior of the nanogel, particularly a rapid sol–gel transition [12,13].

Formulations with concentrations of 5–10/0.1–0.3% remained in a liquid state at both 5 and 30 °C, indicating an absence of thermoresponsive behavior. Conversely, those containing 20/0.1–0.3% remained liquid at 5 °C but exhibited a sol–gel transition at 30 °C within approximately 10 min. The 20/0.1 formulation displayed low viscosity, whereas 20/0.3 resulted in a highly viscous system. The optimal composition, determined as 20/0.2, was selected as the basis for incorporating EO*Pb* into the nG system due to its rapid sol–gel transition properties.

Furthermore, stability tests showed that adding up to 1% EO*Pb* resulted in stable and transparent formulations, with no phase separation observed after the accelerated stability tests. However, concentrations above this threshold led to formulations with low stability, exhibiting creaminess and phase separation [12]. For this study, the F127/974P/EO*Pb* ratio was fixed at 20/0.2/1% (*w*/*w*) and CUR was added at concentrations ranging from 0.01 to 0.05% (*w*/*w*), resulting in the formulations outlined in Table 1. This was accomplished by testing the maximum amount of CUR that could be incorporated into the gel matrix containing 1% (*w*/*w*) EO*Pb*, followed by an evaluation of the system’s accelerated stability. Formulations containing up to 0.03% CUR (nG*P*C1-nG*P*C4) remained stable even after being subjected to accelerated stability testing (seven cycles) and shelf-life testing over 180 days. However, when the CUR content exceeded 0.04%, creaming effect was observed in nanogel nG*P*C5 within 48 h of preparation (Figure 1C). Therefore, the nG*P*C4 formulation, which exhibited the highest percentage of the bioactive EO*Pb*/CUR compounds, was selected for further characterization and referred to throughout the study as nG*P*C.

### 3.2. Characterization of Nanogels

#### 3.2.1. FTIR

Figure 2 displays the FTIR spectra of the nanogels nG, nGP, and nG*P*C. The analysis of the nG spectrum (Figure 2A) reveals a broad and intense absorption band at 3384 cm^−1^, overlapping with the band at 3564 cm^−1^. Due to the miscibility of 0.2% 974P in 20% (*w*/*w*) F127, part of the 974P-974P and F127-F127 intermolecular interactions is replaced by the formation of F127-974P crosslinked hydrogen bonds. The band observed at 3384 cm^−1^ in the nG spectrum is related to a higher-energy O–H stretch, with a blue shift (Δν = 180 cm^−1^). However, a relatively large proportion of F127-F127 hydrogen bonds remain present, as indicated by the strong and broad absorption band at 3564 cm^−1^, which justifies the coexistence of micellar structures of F127 in the nanogel composition. Figure 2B shows the FTIR spectrum of the nG*P* nanogel.

The incorporation of 1% EO*Pb* into the nG matrix induces notable changes in the absorption bands characteristic of the F127/974P system. A prominent and sharp band at 3676 cm^−1^ emerges due to a significant red shift (Δν = 112 cm^−1^) from the original 3564 cm^−1^ band observed in the nG spectrum. This shift suggests a reduction in the energy required for O–H stretching, likely attributed to EO*Pb* molecules disrupting some F127-974P interactions to occupy the pores of the nG*P* matrix. Additionally, the band at 3384 cm^−1^ in nG undergoes a blue shift to 3340 cm^−1^ (Δν = 44 cm^−1^), possibly indicating the formation of new hydrogen bonds between F127/974P and the geranial and neral isomers. However, it is likely that most EO*Pb* molecules preferentially associate with the hydrophobic PPO chains of F127. This hypothesis is reinforced by the higher energy demand for asymmetric C–H stretching, evidenced by the observed blue shift (Δν = 19 cm^−1^) from 2904 cm^−1^ (nG) to 2885 cm^−1^ in the nG*P* spectrum (Figure 2B), suggesting a significant increase in hydrophobic interactions within the system.

Figure 2C,D show the spectra of the nG*P*C material. The intense band related to O–H stretching is split into three signals. The signals at 3550 and 3475 cm^−1^ are attributed to free O–H stretches of CUR molecules and intermolecular hydrogen bonds between F127/974P and CUR molecules, respectively. The signal at 3414 cm^−1^ is attributed to F127/974P intermolecular hydrogen bonds. The presence of the band at 2885 cm^−1^, assigned to C–H stretching, suggests that the highly hydrophobic CUR molecules preferentially associate with the PPO chains of F127.

#### 3.2.2. SEM

SEM is widely recognized as a crucial technique for examining the intricate architectures of nano- and microstructured formulations. Its superior high-resolution imaging capability makes it indispensable for characterizing materials at scales essential to pharmaceutical applications, providing valuable insights into their structural attributes and functional potential. In this study, SEM was utilized to assess both the surface and internal morphology of the lyophilized nG nanogel.

nG micrographs (Figure 3A,B) reveal that the nG nanogel matrix exhibits a porous architecture, featuring interconnected channel-like structures characteristic of crosslinked F127/974P polymer chains due to intermolecular hydrogen bonding. These morphological characteristics offer valuable insight into the properties of nG, suggesting that its porous network effectively retains the liquid phase containing EO*Pb*/CUR molecules, thereby increasing viscosity through strong hydrogen-bonding interactions with water. In fact, the incorporation of EO*Pb*/CUR into nG induces notable morphological modifications in the nanoformulation. SEM micrographs of nG*P*C reveal a rougher surface texture, suggesting that the pores have been filled, as evidenced by their less defined appearance (Figure 3C–F). In agreement with the FTIR findings, the robust intermolecular interactions between EO*Pb*/CUR and the PEO polymer chains restrict the development of pores throughout the lyophilization process.

#### 3.2.3. DLS

The DLS technique is widely used in nanogel characterization due to its ability to accurately measure particle sizes in suspension at the nanoscale [36]. In addition to evaluating the particle size distribution and providing valuable insights into the colloidal stability and intrinsic properties of nanogels, DLS was also fundamental in determining the diffusional thermodynamic parameters of activation. This highlights the crucial role of DLS not only in assessing nanogel stability but also in obtaining key activation parameters that govern their diffusion behavior. This technique is particularly effective in monitoring changes in hydrodynamic radius ($R_{h}$) in response to variations in temperature, pH, and other environmental factors, providing crucial data for developing nanomaterials designed for pharmaceutical and medicinal applications. Analysis of the $R_{h}$ values for nG reveals a marked reduction in particle size with increasing temperature (Table 2).

DLS measurements were performed on solutions diluted to below 1% (m/v). At concentrations lower than the critical micelle concentration (CMC) of F127, the demicellization process is promoted, resulting in a decreased population of smaller particles. This phenomenon arises due to the system’s overall entropy, which drives the aggregation of smaller particles into more polydisperse structures ($R_{h}$ = 661 nm ± 6 and PDI = 0.34). However, since the micellization process of triblock copolymers is highly temperature-dependent, as the temperature of the nG system is increased from 25 to 32 °C, the critical micelle temperature (TMC) compensates for the CMC [37,38,39,40]. This results in the formation of smaller particles with reduced polydispersity ($R_{h}$ = 124 nm ± 5 and PDI = 0.24), indicating an increase in the system’s total entropy and promoting enhanced particle diffusion, as shown in the analysis of the Diffusivity ($D_{if}$) as a function of temperature in Figure 4A. This behavior highlights the thermoresponsive nature of the nanogel. At higher temperatures of 32 and 37 °C, the diffusion processes of nG become even more pronounced, leading to a substantial reduction in nanoparticle sizes ($R_{h}$ = 17 nm ± 3 and PDI = 0.25).

The decrease in $R_{h}$ with increasing temperature may be attributed to the dehydration of the PEO groups in the hydrophilic corona and the PPO core at higher temperatures, which leads to a reduction in $R_{h}$ [40,41]. This process occurs because the rise in thermal energy induces gradual dehydration in non-ionic surfactants, increasing the tendency for separation between the pseudo-phases in the aqueous environment due to shifts in the dynamic equilibrium of micellization. As the temperature increases, the dehydration of both the core and the PEO groups intensifies, causing structural changes in the copolymer. In addition to dehydration, the higher temperature promotes a more efficient entanglement between the PEO and PPO groups, leading to a notable decrease in the size of the diffusing species within the medium [40,42]. Furthermore, the reduction in $R_{h}$ could be related to the fact that copolymers with lower molar mass remain as unimers at elevated temperatures. Triblock architecture copolymers, such as F127, exhibit a dynamic equilibrium between unimers and micelles through an insertion–expulsion mechanism, and between micelles of various geometries and sizes via fusion–fragmentation processes [43].

These characteristics, along with the temperature increase, lead to greater friction between the terminal groups due to water molecule extrusion, ultimately promoting an equilibrium shift toward smaller micelles. As the temperature rises, the F127 copolymer undergoes structural reorganization driven by dehydration and heightened friction between the copolymer’s terminal groups, favoring fragmentation processes and altering the micellization equilibrium [40]. This phenomenon is mainly driven by the copolymer’s structural arrangement in solution, where the hydrophilic PEO segments extend beyond the micelle core, contributing to increased fluidity compared to copolymers with a higher hydrophobic content. Additionally, F127 exhibits anomalies related to macromolecular relaxations, primarily due to the conformational changes of the methyl groups in the PPO blocks. These changes are induced by the extrusion of water molecules from the PPO segments as temperature increases. These unique properties of F127 are responsible for its self-diffusion behavior and characteristic micellization equilibria when compared to more hydrophobic copolymers.

The assessment of $R_{h}$ values for the nG*P*C material demonstrates a distinct behavior when compared to the nG formulation. Unlike nG, the increase in temperature does not significantly reduce the particle size of the nG*P*C material. The observed particle sizes for nG*P*C, ranging from 237 ± 17 nm to 333 ± 22 nm, are consistent with the stability studies. However, the self-organization dynamics of these materials diverge from those of nG, with the inclusion of EO*Pb*/CUR leading to a reduction in diffusion coefficients ($D_{if}$) within the system (Figure 4B). The observed decrease in the $D_{if}$ values for nG*P*C can be attributed to the impact of the EO*Pb*/CUR on the self-organization behavior of the nanogel system. The reduction in the diffusion coefficients reflects shifts in the dynamic micellization equilibria, which are influenced by the multicomponent nature of the system. Additionally, copolymers with concentrations ≤0.05 g/mL exhibit different aggregation equilibria compared to those at concentrations near or above 0.1 g/mL. These changes in aggregation behavior and thermodynamic properties under varying temperature conditions explain the observed variations in the material’s behavior in solution [44].

To gain deeper insights into the effects of temperature, the diffusional activation energy ($E_{ad}$) and the thermodynamic parameters of diffusion ($\Delta H_{d}^{‡}$ and $\Delta S_{d}^{‡}$) were calculated using the Arrhenius law (Equation (Equation 2)) (Figure 4C,D) and Eyring’s theory (Equation (Equation 3)) (Figure 4E,F), both adapted for diffusion processes [33,44,45]. Activation parameters play a critical role in elucidating the mechanisms underlying diffusion, as they provide insights into the energy required for these processes to occur. For the nG system, the calculated value $E_{ad}$ (greater than zero, as shown in Table 3 confirms the temperature dependence of the diffusion process. Specifically, an $E_{ad}$ of 208.53 kJ/mol indicates that the system utilizes thermal energy to facilitate the diffusion of species in solution. This is accompanied by an enthalpy change ($\Delta H_{d}^{‡}$) of 168.20 kJ/mol, signifying energy absorption, and an entropy change ($\Delta S_{d}^{‡}$) of 0.31 kJ/K·mol, reflecting an increase in system disorder.

The temperature rise enhances the hydrophobicity of the copolymer and increases system entropy, leading to the formation of micelles and copolymeric pre-aggregates with smaller hydrodynamic radius ($R_{h}$). This, in turn, increases the excluded volume of the copolymer in solution, which refers to the geometric space occupied by the macromolecular segments. As thermal energy is absorbed, the number of possible configurations available to the system expands, allowing for a broader distribution of species within the nG system and promoting diffusion in the medium. This thermodynamic behavior highlights the adaptability of the system to temperature variations and its potential for temperature-sensitive applications.

The nG*P*C system exhibited a reversal in thermal behavior, with an $E_{ad}$ value of −10.76 kJ/mol, contrasting sharply with the nG system. This shift highlights the impact of incorporating EO*Pb*/CUR, which significantly disrupt the micellization equilibria of the material. At this stage, the thermal energy provided to the system is predominantly utilized in alternative processes, such as structural reorganization, rather than in mass transport via diffusion. This is evident from the exothermic nature of the process, reflected by a $\Delta H_{d}^{‡}$ value of −13.30 kJ/mol. The exothermic nature of the diffusion process reveals two key points. First, incorporating EO*Pb*/CUR introduces strong intermolecular interactions between components, resulting in energy release as the system reorganizes itself. This finding was further supported by FTIR analysis, which revealed that the more hydrophobic EO*Pb* molecules predominantly associate with the PPO segments of F127. This interpretation is reinforced by the observed increase in energy required for the anti-symmetric C–H stretching, as evidenced by the enhanced intensity and blue shift (Δν=19 cm^−1^) of the absorption band, shifting from 2904 cm^−1^ (nG) to 2885 cm^−1^ in the nG*P*C spectrum (Figure 2). This shift suggests a notable enhancement in the system’s hydrophobic interactions, consistent with the enthalpic interaction processes identified.

Second, the decrease in diffusional entropy ($\Delta S_{d}^{‡}=-0.29$ kJ/K·mol) indicates a reduction in the configurational possibilities of the system. As energy is released, the system becomes more ordered, restricting the range of diffusional configurations.

The synergistic effect of these changes is further validated by in vitro applications in cutaneous leishmaniasis. The nG*P*C system demonstrated superior efficacy compared to the control material, underscoring the role of EO*Pb*/CUR in altering the dynamic organization of the nanogel. These incorporations were pivotal in inducing shifts in activation patterns, resulting in an inversion of the diffusional mechanistic behavior. The influence of temperature on the nG and nG*P*C systems is further emphasized by the values of the diffusional Gibbs free energy function ($\Delta G_{d}^{‡}$) as a function of temperature, presented in Table 3, illustrating the thermodynamic distinctions between the two systems.

The increase in temperature for the nG system leads to a progressive decrease in $\Delta G_{d}^{‡}$ values, as shown in Figure 5A. This trend indicates that higher temperatures promote diffusion processes, as less energy is required for the micellar systems to diffuse within the solution. This observation aligns with reductions in $R_{h}$, $\Delta H_{d}^{‡}$, and $\Delta S_{d}^{‡}$, reflecting enhanced micellar dynamics and greater diffusional efficiency at elevated temperatures. In contrast, the nG*P*C system exhibits a thermodynamic inversion with increasing temperature, as previously discussed. This is characterized by negative $E_{ad}$ values and a monotonically increasing trend in $\Delta G_{d}^{‡}$ (Figure 5B). The system also shows a decrease in $\Delta S_{d}^{‡}$, indicating molecular reorganization and an increased ordering of water molecules to solvate monomeric micelles, micellar pre-aggregates, micelles, and EO*Pb*/CUR as temperature rises. The exothermic $\Delta H_{d}^{‡}$ values observed with increasing temperature suggest that the transfer of unimers, micellar pre-aggregates, and EO*Pb* from the solution into the micelle formation is a thermodynamically favorable process. The hydrophobic PPO segments of the copolymer predominantly drive this phenomenon. This complex thermodynamic mechanism explains the monotonically increasing $\Delta H_{d}^{‡}$ values: as the system undergoes self-association and structural reorganization, greater energy is required for these processes to progress.

The diffusion mass transfer profiles of both the nG and nG*P*C systems reveal complex behaviors. The intrinsic properties of F127, combined with the incorporation of EO*Pb*/CUR, amplify the hydrophobic driving forces within the system. These interactions are responsible for the observed structural changes and thermodynamic inversions, further highlighting the intricate relationship between molecular architecture, temperature, and diffusion dynamics.

### 3.3. Rheological Analysis

The flow curve analysis of the nG*P*C sample, evaluated up to a shear rate of 1500 s^−1^ at 5 °C, revealed characteristic Newtonian fluid behavior (Figure 6A). The shear stress versus shear rate curve was linear, starting at the origin, and reaching 250 Pa at 1500 s^−1^. Notably, the forward and backward curves overlapped completely, indicating the absence of hysteresis. This reinforces the rheological stability of the material and rules out thixotropic or rheopectic effects. The constant viscosity, calculated at 0.167 Pa·s, suggests that nG*P*C maintains uniform rheological properties even at low temperatures such as 5 °C. At 25 °C, the rheological analysis of the nG*P*C sample demonstrated non-Newtonian behavior, contrasting with the Newtonian characteristics observed at 5 °C (Figure 6B). The shear stress versus shear rate curve exhibited shear-thinning behavior, where viscosity decreased with increasing shear rate. This indicates an internal fluid structure that reorganizes under high shear forces, a common characteristic of suspensions or nanogels. The forward curve showed a decrease in shear stress from 200 s^−1^ to 1500 s^−1^, suggesting viscoelastic or thixotropic responses, while the backward curve was nearly linear, indicating incomplete structural recovery of the fluid.

At 32 °C, the nG*P*C sample displayed rheological behavior characteristic of non-Newtonian systems, with a pronounced thixotropic pattern (Figure 6C). During the test, the nanogel’s viscosity decreased as the shear rate increased, but it did not fully recover after the removal of shear forces. This suggests that the internal structure of the nanogel reorganizes and becomes more fluid under high-shear conditions. Such behavior is typical of materials where viscosity depends on shear history, as observed in many viscoelastic or thixotropic systems. The forward curve exhibited three peaks followed by drops in viscosity, indicating continuous structural changes within the nanogel as the shear rate increased. However, the structure did not fully restore itself after each peak. This highlights the nanogel’s dynamic response to shear, making it easier to apply at high shear rates but with viscosity not returning to its initial state. On the other hand, the backward curve was almost linear but showed a steeper slope than experiments conducted at 25 °C. This indicates that after shear, the sample remained more fluid, with a partially reorganized structure that did not entirely return to its previous condition.

This rheological behavior has significant implications for the topical applications of nG*P*C, particularly at temperatures ranging from 25 to 32 °C, which correspond to typical human skin temperatures. The reduction in viscosity with increasing shear rate facilitates the application of the nanogel, enhancing its spreadability over the skin. The thixotropic behavior, characterized by variations in shear stress, is advantageous for the controlled release of bioactive compounds, enabling a more effective and gradual therapeutic delivery. This profile is especially desirable in topical treatments, as it ensures prolonged and targeted action. Furthermore, the observed rheological stability suggests that the nanogel nG*P*C (Figure 6D) maintains its efficacy under physiological conditions, underscoring its strong potential for dermatological applications where stable and controlled performance is critical under varying skin temperatures.

### 3.4. In Vitro Assay Against LLa Promastigotes

Leishmaniasis remains a major global health challenge, particularly in tropical and subtropical regions where access to effective treatments is limited. Conventional therapies, such as pentavalent antimonials, AmB, and miltefosine, suffer from significant drawbacks, including high toxicity, elevated costs, prolonged treatment duration, and emerging drug resistance. These limitations highlight the urgent need for novel therapeutic approaches that offer improved efficacy, safety, and accessibility. In this context, the nG*P*C represents a significant advancement by leveraging the synergistic effects of EO*Pb*/CUR within a thermosensitive system. The use of FDA-approved excipients and GC principles enhances its economic viability and sustainability, reducing production costs and improving accessibility. Notably, the nG*P*C is compatible with dPDT, eliminating the need for high-energy artificial light sources and expanding its clinical potential.

To evaluate its therapeutic potential, in vitro assays were conducted to assess its antileishmanial activity against *LLa* promastigotes. The primary objective of this study was to evaluate the therapeutic potential of EO*Pb*/CUR when incorporated into an F127/974P nanogel matrix at low concentrations [26,27,29]. Specifically, the study aimed to determine whether these concentrations would maintain their biological efficacy upon encapsulation, considering that the nanogel matrix could modulate the therapeutic performance of the EO*Pb*/CUR compounds. It is crucial to acknowledge that the excipients used in nanogel preparation may influence the cytotoxic profile of the bioactives. This aspect becomes particularly relevant when analyzing the concentration range of EO*Pb*/CUR within nanogel formulations, as elevated concentrations could enhance cytotoxicity, potentially compromising the safety and therapeutic index of the final formulation.

To test this hypothesis, the efficacy of nanogel formulations, including empty nanogels, those containing CUR, those containing EO*Pb*, and the combination of both compounds, was evaluated. This experimental design allowed for a comprehensive comparison of the effects of each formulation, both in isolation and in combination, thereby providing critical insights into the potential synergistic or additive effects of the EO*Pb*/CUR combination. Furthermore, this setup allowed for the assessment of the influence of the nanogel matrix itself on the bioactivity of the encapsulated compounds. While the encapsulation within the nanogel was hypothesized to reduce the immediate bioavailability of OE*Pb*/CUR due to its controlled release mechanism, it was also expected that the sustained-release properties of the nanogel could provide significant therapeutic advantages. By prolonging the release of active compounds over time, the nanogel system has the potential to enhance overall therapeutic efficacy, particularly for the treatment of chronic or relapsing diseases such as leishmaniasis. This balance between a possible initial reduction in potency and a prolonged therapeutic effect forms the foundation for investigating the effectiveness of these nanogel-based delivery systems.

The effect of the nanogel formulations on *LLa* promastigote cell viability was evaluated at three different time points: 24, 48, and 72 h. Interestingly, the nanogels nG, nGC, and nG*P* showed no anti-promastigote activity at any concentration or time point tested. In contrast, the nanogel nG*P*C demonstrated significant activity, as shown in Figure 7. At the highest concentration tested (17.5 μg/mL of EO*Pb* and 50 μg/mL of CUR), the combination formulation induced over 88% mortality of the promastigotes across all time points (24, 48, and 72 h). This strong effect was observed consistently at each time interval, suggesting a robust and sustained action of the combined compounds. At the lowest concentration tested (2.19 μg/mL of EO*Pb* and 6.25 μg/mL of CUR), the combination formulation still exhibited notable activity, with mortality rates of 24.15%, 21.35%, and 10.40% at 24, 48, and 72 h, respectively. The data also indicate a potential synergistic or additive effect between the two compounds, as the combined formulation exhibited greater efficacy than the individual compounds.

For comparison, AmB (1.95 μg/mL), a standard treatment for leishmaniasis, exhibited mortality rates of 97.67, 96.33, and 96.67% at 24, 48, and 72 h, respectively, indicating sustained high efficacy with minimal variation across time. While the highest concentration of nG*P*C did not reach the mortality levels observed with AmB, it still demonstrated significant activity, particularly considering its advantages in formulation stability, targeted delivery, and reduced toxicity. Furthermore, the progressive decline in mortality rates at the lowest concentration of nG*P*C suggests that sustained drug release or enhanced bioavailability strategies could further optimize its efficacy. Moreover, despite AmB’s well-established effectiveness, its clinical use is limited by high toxicity, particularly nephrotoxicity, necessitating careful monitoring and hospitalization. Additionally, prolonged treatment durations contribute to patient non-compliance and increase the risk of relapse, reinforcing the need for alternative therapies with improved safety and therapeutic profiles [46].

This synergistic effect can be attributed to the complementary properties of the bioactives. EO*Pb*, derived from a scarcely studied species, presents a promising therapeutic approach, while CUR, known for its broad phototherapeutic spectrum, remains underexplored as a leishmanicidal agent in the absence of UV–visible light ($\lambda_{\max}\approx$ 420–430 nm). The dPDT strategy adopted here harnesses the cytotoxic effects of both compounds while eliminating the need for high-energy light sources, thus aligning with GC and sustainability principles [25]. Building upon our previous research, we observed a remarkable improvement in efficacy with the combined formulation. Whereas the EO*Pb*-loaded nanogel previously required 280 μg/mL to achieve ≈ 80% mortality of *LLa* [13], the nG*P*C attained the same effect with just 25/8.75 μg/mL of EO*Pb*/CUR.

Interestingly, the reduction in cell mortality at 72 h in the C3 treatment was also observed with C4, both representing the lowest tested concentrations of the combined compounds. In contrast, the highest concentrations (C1 and C2) exhibited a more pronounced cellular effect, independent of time. This suggests that the potential synergistic effect is primarily concentration-dependent rather than time-dependent, indicating that the drug’s efficacy does not progressively increase over time. This phenomenon may be attributed to a reduction in the availability of compounds due to cellular uptake, highlighting the need for an optimal minimal effective dose. According to Kuti (2016), antimicrobial activity depends on concentration, minimal effective concentration, and exposure time [47]. When the concentration effect is more significant than the time effect, the drug is classified as concentration-dependent, with efficacy determined by the ratio of maximum free drug concentration to the minimal effective concentration. Conversely, when time predominates, efficacy depends on how long the drug concentration remains above the minimal effective dose. Given that nG*P*C demonstrated sustained activity over time at its highest concentration but a progressive reduction at lower concentrations, its efficacy likely follows a concentration-dependent profile. This underscores the importance of optimizing drug delivery to maintain effective concentrations for prolonged periods. Further studies on the mechanism of action are necessary to better elucidate this phenomenon.

Since the experimental conditions of this study included the application of dPDT, the findings strongly support its role in enhancing the therapeutic efficacy of the nG*P*C system. The use of dPDT probably increased the observed leishmanicidal effects, reinforcing the importance of photodynamic activation in this therapeutic strategy. Given that CUR is a potent photosensitizer, its incorporation into the nanogel system alongside EO*Pb* under dPDT conditions may provide a synergistic effect, enhancing the leishmanicidal potential of nG*P*C. These results underscore the relevance of dPDT as an integral component of this approach, suggesting that future optimizations should continue to explore its role in maximizing the therapeutic potential of nanogel-based treatments for leishmaniasis.

## 4. Conclusions

This study demonstrates the potential of the nanogel formulation nG*P*C incorporating EO*Pb*/CUR as a promising therapeutic strategy against *LLa* promastigotes under dPDT conditions. The physicochemical characterization confirmed the stability and thermosensitivity of nG*P*C, ensuring its suitability for topical application. Rheological analysis revealed a transition from Newtonian to shear-thinning and thixotropic behavior, facilitating its application in topical treatment. Spectroscopic analyses confirmed molecular interactions that facilitated improved diffusional properties. Furthermore, DLS measurements revealed that EO*Pb*/CUR incorporation influenced R_*h*_ and thermodynamic parameters, indicating strong intermolecular interactions. The negative diffusional activation energy and entropy suggested enhanced system stability and structural reorganization rather than mere diffusion, reinforcing the efficacy of the formulation. These findings underscore the importance of DLS as a powerful tool for evaluating the synergistic behavior of EO*Pb*/CUR in nanogels. Biological evaluations demonstrated that nG*P*C significantly enhanced anti-promastigote activity compared to free EO*Pb* or CUR, with over 88% mortality at the highest tested concentration and a dose-dependent response at lower concentrations. These findings suggest a synergistic interaction between EO*Pb* and CUR, likely mediated by the nanogel matrix, which improves bioavailability and therapeutic efficacy. Furthermore, the experimental conditions confirmed the relevance of dPDT in enhancing the leishmanicidal effects of nG*P*C, reinforcing its potential as a light-activated therapeutic approach.

The results underscore the potential of nG*P*C in overcoming the limitations of conventional leishmaniasis treatments, particularly by enhancing drug stability, reducing toxicity, and enabling controlled release. Future studies should focus on elucidating the molecular mechanisms underlying the observed synergism, as well as evaluating the impact of dPDT to further enhance therapeutic outcomes. Overall, nG*P*C represents an innovative and effective platform for delivering bioactive compounds against*LLa* promastigotes, reinforcing the relevance of nanotechnology and natural product-based therapies in addressing neglected tropical diseases.

## Figures and Tables

**Figure 1 pharmaceutics-17-00286-f001:**
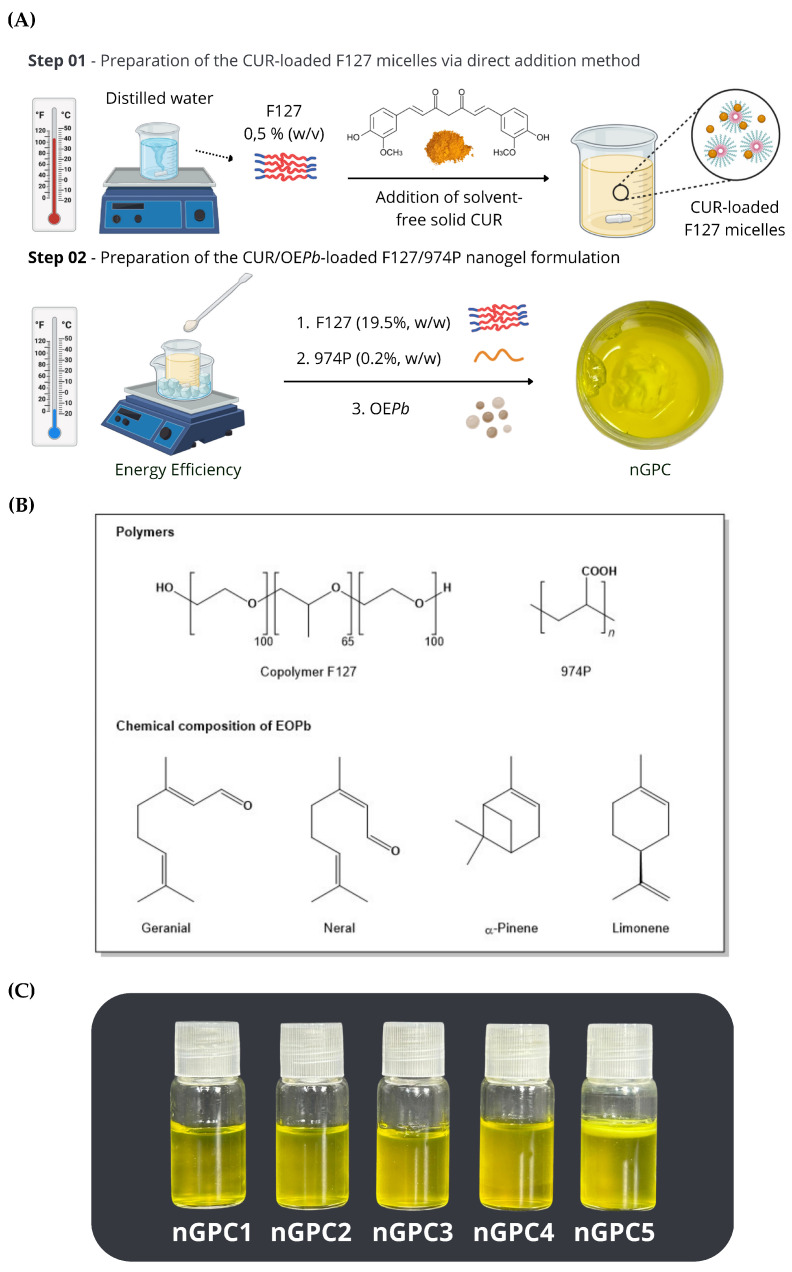
(**A**) Stages of preparation of nG*P*C (nanogel loaded with EO*Pb* and CUR), involving the addition of curcumin (CUR) to the solution under controlled temperature conditions to ensure its uniform dispersion within the polymeric matrix, followed by the incorporation of *Pectis brevipedunculata* essential oil (EO*Pb*). (**B**) Molecular structure of the polymers utilized in nanogel formulation and primary chemical constituents composing EO*Pb*. (**C**) Photographs of the nG*P*C1–nG*P*C5 formulations following the accelerated stability test, where the nG*P*C5 nanogel exhibits a reaming effect.

**Figure 2 pharmaceutics-17-00286-f002:**
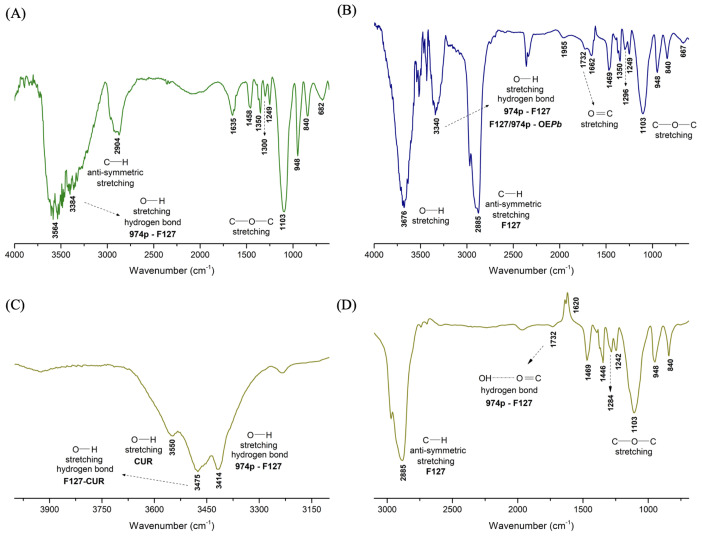
FTIR spectra of the nanogels: (**A**) nG (empty nanogel), (**B**) nGP (EO*Pb*-loaded nanogel) (**C**) nG*P*C (EO*Pb*/CUR-loaded nanogel) in the range of 4000–3100 cm^−1^, and (**D**) nG*P*C in the range of 3100–690 cm^−1^.

**Figure 3 pharmaceutics-17-00286-f003:**
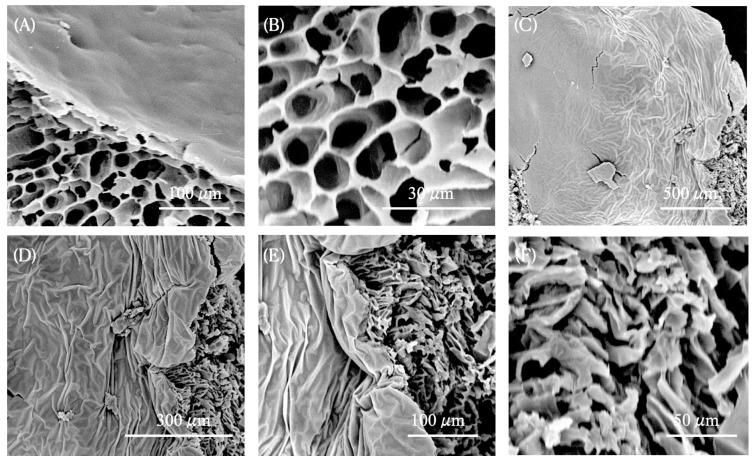
SEM micrographs of nanogels after the freeze-drying process: (**A**) and (**B**) nG (empty nanogel) at magnifications of 1000× and 5000×, respectively; (**C**), (**D**), (**E**), and (**F**) nG*P*C (EO*Pb*-loaded nanogel) at magnifications of 200×, 500×, 1000×, and 2000×, respectively.

**Figure 4 pharmaceutics-17-00286-f004:**
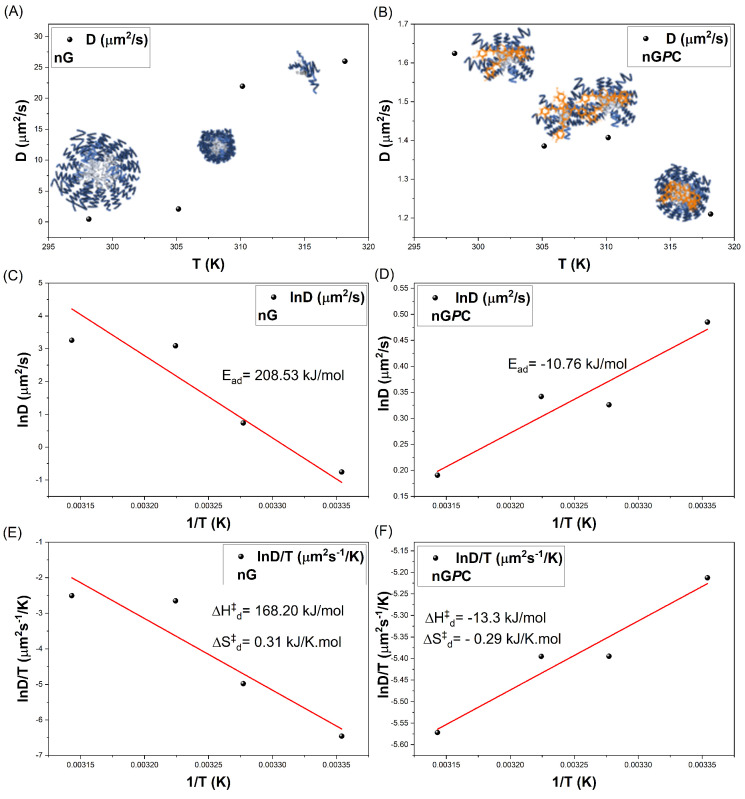
Comparative analysis of the diffusion coefficient D_*if*_ as a function of temperature (K): (**A**,**C**,**E**) nG (empty nanogel) and (**B**,**D**,**F**) nG*P*C (EO*Pb*/CUR-loaded nanogel).

**Figure 5 pharmaceutics-17-00286-f005:**
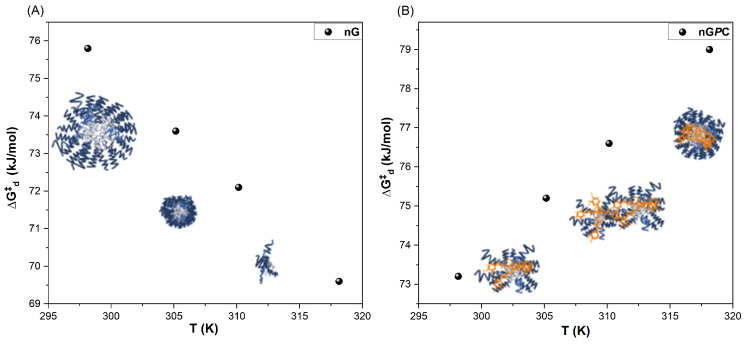
Plot of $\Delta G_{d}^{‡}$ as a function of temperature for the (**A**) nG (empty nanogel) and (**B**) nG*P*C (EO*Pb*/CUR-loaded nanogel).

**Figure 6 pharmaceutics-17-00286-f006:**
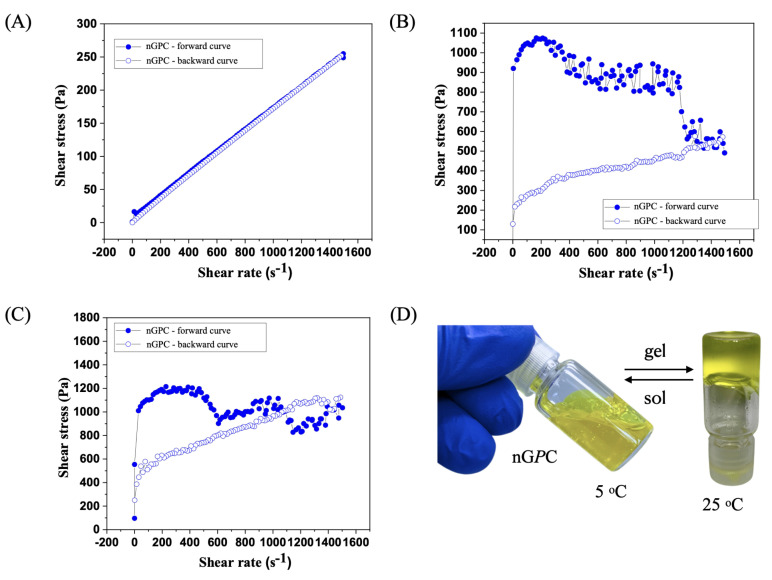
Rheological behavior of the nG*P*C (EO*Pb*/CUR-loaded nanogel) sample at varying temperatures: (**A**) Newtonian fluid behavior at 5 °C with a linear shear stress versus shear rate curve; (**B**) shear-thinning behavior at 25 °C, demonstrating viscosity decrease with increasing shear rate; (**C**) pronounced thixotropic pattern at 32 °C, with viscosity reduction and incomplete structural recovery after shear. The curves demonstrate the nanogel’s dynamic response to shear forces, highlighting its suitability for topical applications; (**D**) real image of the nG*P*C nanogel during the sol–gel transition, showing sol at 5 °C and gel at 25 °C. The curves demonstrate the nanogel’s dynamic response to shear forces, highlighting its suitability for topical applications.

**Figure 7 pharmaceutics-17-00286-f007:**
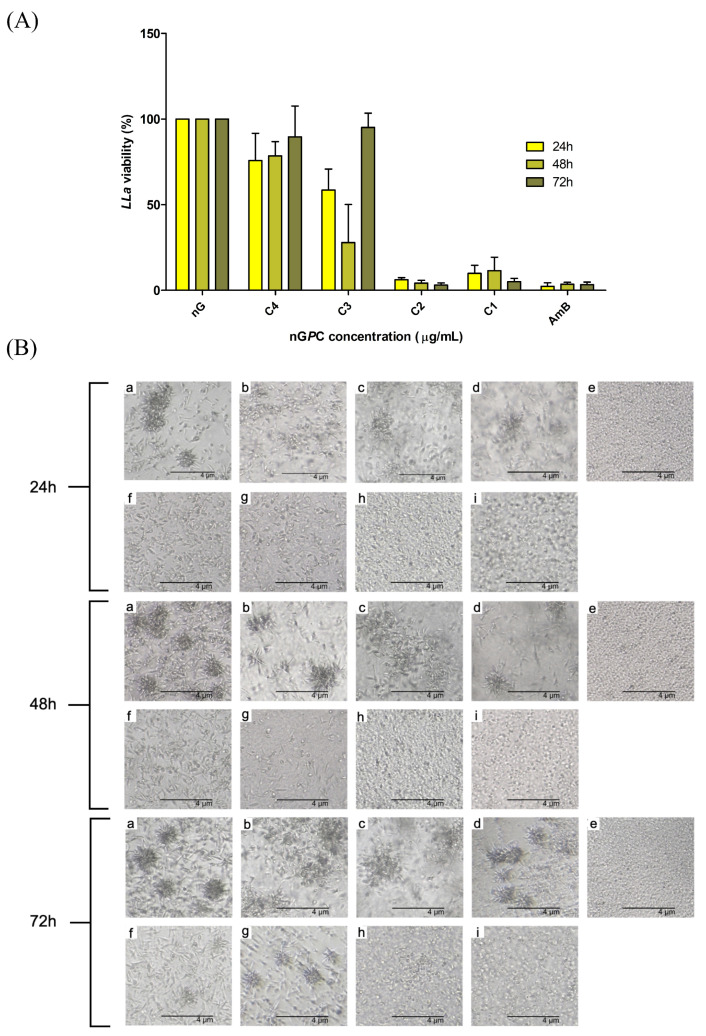
(**A**) in vitro viability (%) of *LLa* promastigote cells in response to different concentrations of nG*P*C (EO*Pb*/CUR-loaded nanogel). The concentrations C4–C1 represent the combination of EO*Pb*/CUR as follows: C4 = 6.65/2.19, C3 = 12.5/4.38, C2 = 25/8.75 and C1 = 50/17.5 (μg/mL). The nG (empty nanogel) and AmB (amphotericin B) were used as negative and positive controls, respectively. nG*P* (EO*Pb*-loaded nanogel) and nG*C* (CUR-loaded nanogel) did not exhibit cytotoxicity against *LLa* at any of the tested concentrations; therefore, their results were not included in the graph. (**B**) Representative images of *LLa* cultures treated with nanogel formulations at different time points: (a) untreated; (b) nG; (c) nGC (50 μg/mL); (d) nG*P* (17.5 μg/mL); (e) AmB (1.95 μg/mL); (f–i) nG*P*C (C4–C1).

**Table 1 pharmaceutics-17-00286-t001:** Optimized nanogel formulations (nG*P*C1–nG*P*C5) with varying CUR and water (% *w*/*w*) concentrations, combined with a fixed composition of F127, 974P, and EO*Pb*.

	Component % (*w*/*w*)					
**Code**	**Water**	**F127**	**974P**	**EO** * **Pb** *	**CUR**	**Stability ^a^**
nG*P*C 1	78.99	20	0.2	1	0.01	S
nG*P*C 2	78.78	20	0.2	1	0.02	S
nG*P*C 3	78.47	20	0.2	1	0.03	S
nG*P*C 4	78.36	20	0.2	1	0.04	S
nG*P*C 5	78.25	20	0.2	1	0.05	C

^a^ S: Stable; C: Creaming.

**Table 2 pharmaceutics-17-00286-t002:** DLS measurements as a function of temperature for the nG (empty nanogel) and nG*P*C (EO*Pb*/CUR-loaded nanogel). Measurements were conducted in quintuplicate, and results are expressed as mean ± SD.

	nG	nG*P*C
Temperature (°C)	R_*h*_ (nm)/PDI	
25	661.00 ± 6.00/0.34	288.00 ± 74.00/0.29
32	124.00 ± 5.00/0.24	296.00 ± 24.00/0.44
37	17.00 ± 3.00/0.25	237.00 ± 17.00/0.42
45	16.93 ± 3.00/0.26	333.50 ± 22.00/0.45

**Table 3 pharmaceutics-17-00286-t003:** Thermodynamic parameters of diffusional process for nG and nG*P*C.

nG		nG*P*C	
Parameter	Value	Parameter	Value
$E_{\mathrm{ad}}$ (kJ mol^−1^)	208.53	$E_{\mathrm{ad}}$ (kJ mol^−1^)	−10.76
$\Delta S_{d}^{‡}$ (kJ K^−1^ mol^−1^)	0.31	$\Delta G_{d}^{‡}$ (kJ K^−1^ mol^−1^)	−0.29
$\Delta H_{d}^{‡}$ (kJ mol^−1^)	168.20	$\Delta H_{d}^{‡}$ (kJ mol^−1^)	−13.30
$\Delta G_{d}^{‡}$ (kJ mol^−1^) at different temperatures			
Temperature (K)	nG	Temperature (K)	nG*P*C
298.15	75.80	298.15	73.20
303.15	73.60	303.15	75.20
305.15	72.10	305.15	76.60
310.15	69.60	310.15	79.00

## Data Availability

The original contributions presented in this study are included in the article. Further inquiries can be directed to the corresponding author.

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
