# Peer review of "Enhanced Nanogel Formulation Combining the Natural Photosensitizer Curcumin and Pectis brevipedunculata (Asteraceae) Essential Oil for Synergistic Daylight Photodynamic Therapy in Leishmaniasis Treatment"

_pharmaceutics, 2025, doi:10.3390/pharmaceutics17030286_

Round 1
Reviewer 1 Report
Comments and Suggestions for Authors
The manuscript "Enhanced Nanogel Formulation of Curcumin with Pectis Brevipedunculata Essential Oil for Enhanced Leishmaniasis Treatment" describes the development of a novel nanogel formulation that incorporates curcumin and the essential oil of Pectis brevipedunculata for the treatment of leishmaniasis. Although the study demonstrates improved anti-promastigote activity in vitro, the results do not meet the standards for publication in a Q1 journal such as Pharmaceutics.
The proposed approaches, such as the use of nanogels for drug delivery and the investigation of synergistic effects between natural compounds, are well-established in the field and do not demonstrate sufficient novelty to warrant publication in a high-impact journal.
The manuscript provides a detailed description of the nanogel formulation and its physical and chemical properties. However, the manuscript does not convincingly demonstrate the novelty and significance of the contribution. Although the synergistic effect between curcumin and P. brevipedunculata essential oil is interesting, it is not sufficiently innovative. Many studies have investigated synergistic combinations of natural substances for treating various illnesses. The manuscript should clearly state what new information or approach it contributes beyond the current literature. This would require a more in-depth analysis of the current body of research and a clear explanation of how this research significantly advances the field.
The potential for photodynamic therapy (PDT) has been mentioned, but it has not been explored experimentally, which weakens the overall impact of the study. Furthermore, a major limitation is the lack of in vivo data. While the in vitro results are promising, they cannot alone justify the publication of this study in a high-impact journal. To assess the translational potential of the nanogel formulation, it is crucial to demonstrate its efficacy in a relevant animal model.
This manuscript requires significant revision before it can be considered for publication.The authors should clearly define the novel aspect of their research: What specific and unique contribution does their study offer compared to the existing literature on nanogel formulations and synergistic therapies for leishmaniasis? This aspect should be clearly stated in the introduction and discussed throughout the manuscript.
The significance of this work should be clearly highlighted. The global impact of the research should be emphasized and the limitations of existing leishmaniasis treatment methods should be addressed. Potential advantages in terms of effectiveness, safety, and cost-effectiveness should be discussed.
In vivo studies (if possible) in a suitable animal model are necessary to support in vitro findings and provide evidence for the proposed approach. The potential synergy between the proposed treatment and photodynamic therapy (PDT) should be explored experimentally rather than simply mentioned as future work. A deeper understanding of the mechanism underlying the synergistic effect should be gained through further research.
It is also necessary to reduce the level of plagiarism from the articles of the authors themselves, especially in the methods section.
In my opinion, reconsideration after major revisions (substantial revisions to text or experimental methods needed) should be carried out.
Author Response
We would like to express our sincere gratitude to the reviewers for their insightful comments regarding our work, pharmaceutics-3491702. It is a pleasure to have the expertise of the reviewers, who will be instrumental in enhancing the quality of the final version of the manuscript. Your valuable feedback is greatly appreciated and will significantly contribute to the improvement of our research and its reception by readers.
Please note that the corrections are highlighted in blue in the text, and our point-by-point responses to the reviewers' comments are detailed in the attached PDF file.

Reviewer 2 Report
Comments and Suggestions for Authors
This study presents a novel approach to “Enhanced Nanogel Formulation Combining the Natural Photosensitizer Curcumin and Pectis brevipedunculata (Asteraceae) Essential Oil for Synergistic Leishmaniasis Therapy.” However, further discussion on the practical application basis could enhance the strength of your paper:
Suggestions
Major comments
- In the introduction section, the rationale for combining curcumin (CUR) and Pectis brevipedunculata essential oil (OEPb) in a nanogel is scientifically sound, but it lacks direct comparison with alternative drug delivery systems such as liposomes, micelles, or nanoemulsions. Please highlight more why was a nanogel chosen over these.
- The stability studies lacked quantitative data on phase separation, particle aggregation, or degradation over time. Did any changes occur in drug content, pH, viscosity, or particle size over the storage period? Please add quantitative data, which can clarify sample stability.
- Figure 7, the treatment by C3 decreased the cell mortality (%) at 72 h compared to 48 h. Please clarify these results.
- The author has provided zeta size and zeta potential of nanogels; however, zeta potential provides a critical idea regarding colloidal stability. Please add the zeta potentials result.
- Highlight more regarding the use of concentration or the ratio between CUR and OEpb for designing formulation.
- During in vitro assay against LLa, a combinational form of CUR and OEpb was performed in last by using DMSO as a vehicle. Could you clarify the % of DMSO?
- Please provide the reference and selected concentration as per reference (Line 501-502, Section 3.4).
- I would recommend adding your blank vehicle to further clarify the real cytotoxicity effects of CUR and OEpb at respective dilutions.
- To represent the synergism effects, use fixed treatment concentration for example, (CUR: OEpb, 1:0, 1:0.25, 1:0.5, 1:0.75, 1:1, 0.75:1, 0.5:1, 0.25:1, 0:1). As for 20 µg/mL overall drug concentration with 1:0.25 ratio between CUR and OEpb, (16 µg/mL CUR + 4 µg/mL OEpb). You can use multiple overall concentrations based on your experience.
- In your formulation table 1, there was 1:0.05 provides phase separation, however, you have used a greater ratio in cell viability. Was your samples stable in this case? Please clarify it.
- Additionally, comparing the cell viability effects between your formulations and standard treatments can improve the impact of the paper.
Minor comments:
- Revise statistical reporting (P-values, IC50)
- Improve the figure resolution
- On each figure or Table legend, please add details about the abbreviation
Author Response
We would like to express our sincere gratitude to the reviewers for their insightful comments regarding our work, pharmaceutics-3491702. It is a pleasure to have the expertise of the reviewers, who will be instrumental in enhancing the quality of the final version of the manuscript. Your valuable feedback is greatly appreciated and will significantly contribute to the improvement of our research and its reception by readers.
Please note that the corrections are highlighted in red in the text, and our point-by-point responses to the reviewers' comments are detailed in the attached PDF file.

Round 2
Reviewer 1 Report
Comments and Suggestions for Authors
The revised manuscript, entitled «Enhanced Nanogel Formulation Combining the Natural Photosensitizer Curcumin and Pectis brevipedunculata (Asteraceae) Essential Oil for Synergistic Daylight Photodynamic Therapy in Leishmaniasis Treatment», has undergone significant improvement and is now ready for publication in the journal «Pharmaceutics». The authors have effectively addressed the comments of the previous reviewers, resulting in a more concise and impactful presentation of their research.
In particular, the focus of the manuscript has been sharpened. The emphasis is now clearly placed on the synergistic effect achieved through the combination of curcumin with the essential oil of P. brevipedunculata in the nanogel formulation for photodynamic therapy using daylight. The revisions have improved the readability and flow of the text, making the presentation of data and conclusions more clear and accessible. Any questions regarding the design of the experiment and methodology have been satisfactorily addressed.
The current in vitro data serves as a solid foundation for further exploration, but the manuscript would undoubtedly benefit from more extensive in vitro studies that encompass a wider range of cell lines and experimental conditions.
Moreover, in vivo investigations are crucial for translating these promising results into a pre-clinical setting and validating the therapeutic potential of the novel nanogel formulation. The authors acknowledge these limitations and outline plans for future investigations both in vitro and in vivo. Despite these limitations, the manuscript is deemed suitable for publication as it stands.
These future studies will undoubtedly expand upon the current research, providing a more comprehensive understanding of the efficacy and promise of this innovative approach to leishmaniasis treatment.
Author Response
Dear Reviewer,
We sincerely appreciate your thoughtful and constructive comments on our revised manuscript. Your feedback has been invaluable in refining our work, and we are grateful for your recognition of the improvements made.
We are pleased that you find the manuscript more concise and impactful, with a clearer emphasis on the synergistic effect of curcumin and Pectis brevipedunculata essential oil in our nanogel formulation for daylight photodynamic therapy. Your acknowledgment of the enhanced readability and the clarity of data presentation is truly encouraging.
We fully agree with your suggestion regarding the need for further in vitro studies with a broader range of cell lines and experimental conditions, as well as in vivo investigations to validate the therapeutic potential of our formulation in a pre-clinical setting. As mentioned in the manuscript, we recognize these limitations and have outlined our plans for future research in these areas.
Once again, we sincerely appreciate your time and expertise in reviewing our work. Your constructive insights have greatly contributed to strengthening our manuscript, and we are grateful for the opportunity to share our research in Pharmaceutics.
Reviewer 2 Report
Comments and Suggestions for Authors
Thank you for resolving all the comments. However, I would like you to requesuest to apply the following suggestion for further improvement of the manuscripts:
- Please incorporate the drug content present in the nanogel along with the entrapment ratio along with the release profile of drugs.
- Please check line 275 and mention clearly about the table.
Author Response
Dear Reviewer,
We sincerely appreciate all your thoughtful comments and suggestions, which have greatly contributed to improving the quality of our manuscript. Thank you for your time, effort, and valuable insights.
Comment 1
Please incorporate the drug content present in the nanogel along with the entrapment ratio along with the release profile of drugs.
Response 1
Thank you for your valuable suggestion. While we did not perform direct encapsulation efficiency assays, we can reasonably infer that the encapsulation efficiency is close to 100%, as the entire amount of CUR/OEPb was solubilized during the preparation of nanogels nGPC1–nGPC4 (Table 1), and these formulations remained stable after accelerated stability tests. Furthermore, due to the highly hydrophobic nature of the active compounds and their strong interactions with the polymeric chains, we encountered challenges in evaluating the in vitro release profile of CUR/OEPb using Franz diffusion cells. We believe that drug release occurs in conjunction with the polymer matrix as the nanogel begins to dissolve in the biological medium. However, to gain a more comprehensive understanding of the release behavior, we are collaborating with the Department of Biophysics at the Escola Paulista de Medicina, UNIFESP, under the coordination of Prof. Daniele Ribeiro de Araujo. The findings from this investigation will be presented alongside our in vivo studies in a forthcoming publication.
Comment - 2
Please check line 275 and mention clearly about the table.
Response - 2
The citation of Table 01 has been corrected in the manuscript and highlighted in yellow in the revised version.